# The Role of Vegetation on the Dynamics of Water and Fire in the Cerrado Ecosystems: Implications for Management and Conservation

**DOI:** 10.3390/plants9121803

**Published:** 2020-12-18

**Authors:** Carlos A. Klink, Margarete N. Sato, Giovanna G. Cordeiro, Maria Inês M. Ramos

**Affiliations:** Department of Ecology, University of Brasilia (UnB), 70910-900 Brasília, Brazil; nsatoecologia@gmail.com (M.N.S.); giogomes.2402@gmail.com (G.G.C.); nesita03@gmail.com (M.I.M.R.)

**Keywords:** ecosystem services, land use and conservation, public policies

## Abstract

The Cerrado is the richest savanna and is undergoing one of the planet’s most rapid land transformations for pasture and agriculture; around 45% of the biome has been deforested. Agriculture is of strategic importance to Brazil, but it also modifies ecosystems and jeopardizes habitats and biodiversity. Well-managed agricultural lands can have a favorable impact on environmental conservation. In this paper, we reviewed our current knowledge about water ecology and fire management to show that an ecosystem services perspective can bring about a conciliation of agriculture production with conservation by supporting effective land use decision-making and the optimization of public policy. The landscape/watershed scale seems to be the most relevant for decision-making on how to achieve production and conservation results. This scale appears to be an appropriate level for engaging with stakeholders. Fire frequency and timing (season) combination are determinant of individuals’ survivorship. The combination determines vegetation recovery, and it is important to maintain high biodiversity, especially for the herbaceous layer, but it is a limitation to woody vegetation recovery. A pragmatic and conciliatory land use agenda must be based on scientific knowledge and support innovative decision-making solutions for policy-makers and stakeholders, particularly farmers and donors.

## 1. Introduction

In Brazil and abroad, the Cerrado biome is seen as one of the last major land frontiers of the world. In contrast, it also harbors high biodiversity, dynamic landscapes, and is critical to protecting water resources, as it hosts the headwaters of important Brazilian hydrographic basins which supply nearly all the nation’s biomes.

It has become Brazil’s largest source of soybeans and pastureland, and a significant producer of corn, cotton, and sugarcane. Land use decisions are driven by technological innovations, capital investments, knowledge, and policy interventions. The primary objective of Cerrado land use is to expand intensive agriculture to meet the demand for agricultural products in Brazil and internationally. Pastures cultivated with African grasses, cash crops and planted forests are the main types of land use that today cover 40% of the Cerrado; native vegetation still covers 56% of the biome [1,2].

The strong performance of the Brazilian economy during the 1960s and 1970s, associated with a national development policy aimed at integrating the “empty” spaces of the Cerrado and Amazon biomes into the Brazilian economy, created the business environment for investments. Policies aimed at expanding the agricultural output of the Cerrado included subsidies, developmental programs, and the use of new technologies [3,4,5,6,7].

From a revenue-generation perspective, the benefits of commercial agriculture in the Cerrado are a success: soybeans and soy products are among the largest of Brazil’s export commodities, and it supports the largest cattle herd in the country. Even so, the development of modern agriculture in the region has exacerbated social inequality at a high environmental cost, leading to landscape fragmentation, loss of biodiversity, biological invasion, soil erosion, water pollution, land degradation, heavy use of chemicals, changes in fire regime, and alterations of carbon and water cycling. Transformation of the landscape continues, but a series of new policies and instruments are being implemented that may change the future direction of land use in the Cerrado.

Several environmental and economic conditions have favored these transformations. Although rainfall distribution throughout the year is uneven, the mean total annual rainfall is considered sufficient for crop production. Sunshine is year around and mild temperatures do not restrict photosynthesis. The Cerrado’s level topography suits mechanization, and the savannas are cheaply and more easily cleared for farming or cattle ranching than tropical rainforest.

The Cerrado climate is mostly rainy and tropical with two well-defined seasons, the wet (rainy) season, which usually runs from October through May, and the dry season from April through September. The average rainfall in the central Cerrado is 1500 mm per year, but there is a large variability in precipitation that can range from 400 mm (in the transition areas with the Caatinga biome in the Northeast of Brazil) to 2400 mm of rain per year (in the transition areas towards the Amazon rainforest). It is common to have dry spells of up to 20 days without rain during the wet season (locally known as “veranicos”), usually between December and February [8]. At the peak of the dry season it is common to have long periods without any rain (August–September).

In this paper, we reviewed our current knowledge about water ecology and fire management to show that an ecosystem services perspective can bring about a conciliation of agriculture production with conservation by supporting effective land use decision-making and the optimization of public policy. Both the conservation of ecosystems and agriculture are of strategic importance to Brazil. We argue that the richness of Brazilian policies on land use can both provide well-managed agricultural lands and have a favorable impact on environmental conservation.

## 2. Water in the Cerrado Biome

Water is the most limiting resource for plant productivity in several terrestrial ecosystems. The seasonality in soil water availability is considered a key factor in determining ecosystem function and vegetation structure [9]. Plant and microbial physiological activity are highly dependent upon water availability, and therefore, so are ecosystem processes such as primary production and decomposition. In the Cerrado biome, ecosystem processes that are strongly responsive to soil moisture include vegetation phenology [10], grass biomass production, carbon assimilation rates [11,12] and soil respiration [13].

The central location of the Cerrado biome in the Brazilian territory, its high elevation, and its deep soils are critical to protecting water resources, as they support a wide network of rivers and aquifers that supply important hydrographic basins of Brazil (Figure 1). This abundance commonly raises debates regarding the role of Cerrado in providing water to other regions of Brazil, including more than 50% of the country’s hydroelectric production [14], and in maintaining water quality, which is related to the conservation of riparian vegetation alongside rivers [15,16,17,18,19,20].

It is well documented that the ecological heterogeneity in the Cerrado (Figure 2) results from the variation in climate (rainfall and temperature), geology, geomorphology, soil, and dominant vegetation type [8,22,23]. This ecological heterogeneity influences both the energy (input and output of energy flux) and the water balance (inputs and outputs of water through the vegetation and the soil), which depends on hydrological processes such as precipitation, evapotranspiration, water interception by vegetation, stem flow, water infiltration and percolation in the soil profile, soil surface runoff, and water absorption by roots.

### 2.1. Changes in Ecosystem Vegetation Cover Affect Hydrology and Precipitation

For a long time, scientists and practitioners have been debating the effects of vegetation cover on rainfall and how this debate should inform policy making, especially in the tropics [24,25,26,27,28,29,30,31,32,33]. It is common to hear anecdotes from farmers that after the suppression of native savannas, “downstream water comes with force.” However, deforestation usually affects hydrological processes, changing runoff characteristics, water yield, groundwater recharge, and soil infiltration capacity, as well as precipitation and evaporation rates [16,34,35,36,37,38].

Scientific evidence shows that increases in streamflow are typically short-lived and can ultimately degrade water quality and increase vulnerability to flooding [39]. These changes are especially worrying when the Cerrado’s natural vegetation is replaced by pasture and croplands that are poorly managed. Therefore, establishing sustainable agricultural systems that prevent soil degradation can provide important environmental benefits in the Cerrado region, including reduced erosion rates, and increased nutrient uptake, water storage, carbon sequestration and biodiversity, among others [40,41,42,43,44].

The current debate revolves around whether forests are net users or producers of water. The “supply-side” thinkers argue that forests are net producers of waters and increase rainfall at regional scales. On the other hand, the “demand-side” thinkers argue that trees are users of water and that they decrease water availability for other users (farmers and city dwellers) at the catchment scale [45]. This debate on the role of forests in controlling water balance has been on for over a century. The issue of forest–rainfall feedback has gained prominence for the past 30 years due to concerns related to deforestation and climate change in the policy and scientific domains, and apparently the controversy is still not resolved [45,46,47,48].

In Brazil, this debate is heated by concerns about the significant rates of deforestation in the country. The deforestation of forests and savannas for the expansion of agroecosystems is the main driver of land cover changes in the Brazilian territory [49,50,51,52,53,54,55,56,57,58,59,60,61]. The conversion of natural ecosystems releases carbon to the atmosphere and affects land surface biophysical properties, including evapotranspiration, albedo (the reflectivity of the land surface), and the roughness of the vegetation, all of which influence local precipitation patterns [25]. The release of aerosols from burning forests and savannas can also affect the formation of clouds and precipitation [25,62]. All these changes have implications for agricultural production, which depends on temporal and spatial patterns of temperature and precipitation.

Climate and ecological modeling as well as field studies have shown some influence of the deforestation of tropical forests on the formation of clouds and rainfall. Modeling projections performed for the National Plan for Adaptation to Climate Change published predictions that the Cerrado might experience a reduction in precipitation of 35–45% by the end of the century, and that the regional temperature might increase by up to 5–5.5 °C [63]. Costa and Pires [64] have concluded that Cerrado deforestation may increase the duration of the dry season from 5 to 6 months. They performed a modeling analysis for Cerrado’s future scenarios, based on atmospheric and biophysical parameters, and related the effect of deforestation to rainfall and the duration of the dry season.

More recently, Cohn et al. [65] explored non-local warming within Brazil’s Amazon and Cerrado biomes, using two datasets, one consisting of in-situ air temperature observations and a second consisting of remotely-sensed observations of land surface temperature. The study evaluated air temperature response to non-local forest loss occurring at distances from 1 to 60 km from the monitoring sites. The results showed substantial non-local warming, suggesting that historical evaluations underestimate or misattribute warming to local change, where non-local changes also influence the patterns of temperature increases.

Transforming native ecosystems into pasture and croplands has induced regional climate change in the past 40 years, due to reduced net surface radiation and evapotranspiration, and increased sensible heat flux and land surface temperature after deforestation [66]. The authors reported that the Xingu Indigenous Park is 3 °C cooler than the mosaic of agricultural lands and the fragmented forests that surround the park in the state of Mato Grosso. They also reported reduced rainfall and extended dry season in deforested areas in the state of Rondônia.

More specific analyses have predicted significant differences in productivity, carbon, and evapotranspiration rates when native Cerrado ecosystems are compared to the most important agroecosystems of the region, pasture, and croplands. Using “vegetation greenness indexes” and evapotranspiration indexes derived from remote sensing (MODIS) for the 2000–2012 period, Arantes et al. [67] categorized the effects of land cover and land use changes into primary productivity, carbon, and evaporative fluxes of these different ecosystems. The authors found that for the year 2002, the total amount of carbon biomass for the entire Cerrado region (in the wet season) was estimated as 28 Gt of carbon, and evapotranspiration was 1336 Gt of water. From this total, native ecosystems accounted for 15 Gt of carbon and 760 Gt of water, while pastures and croplands combined summed up to 12 Gt of carbon and 576 Gt of water. Simulating future land cover scenarios (for the year 2050), the authors reported that evapotranspiration from the remaining native Cerrado ecosystems (predicting current trends of deforestation) would be 394 Gt less than in 2002.

Clearly, there are some scientific predictions for the possible effects of changes in vegetation cover and rainfall, but we still lack more definitive empirical evidence to fully understand and make policy-relevant predictions for the ecosystems–rainfall relationship. There is also a lack of evidence about the viability of the concept of ecosystem services for decision-making in the Cerrado, which is discussed below.

### 2.2. The Role of Water as an Important Ecosystem Service in the Cerrado

A number of initiatives have been launched with the goal of mainstreaming the concept of ecosystem services in Brazil, for public policy, business [68], and public awareness [69]. All of these have been important for engaging academia, civil society, and the media, but they still lack the necessary traction to make ecosystem services a more operational concept for decision-making and investments in Brazil. The empirical, scientific, or policy-relevant literature on the importance of water as an ecosystem service in the Cerrado is scarce.

There has been an increase in technical and methodological ecosystem valuation studies in Brazil, but most were site-specific and did not evaluate the value of ecosystems at larger scales or the construction of an environmental services market, particularly related to rural development in Brazil [70]. More recently, the use of other economic instruments, such as water charges and payments for ecosystem services, started to be introduced. The National Water agency launched in 2011 the water producer program, to financially compensate farmers that invest in soil and water conservation [71,72].

This national program is one of the instruments of the national water policy and seeks to promote the sustainable use and conservation of water resources. It is targeted toward the watershed level and requires the participation of both public and private entities. The goal is to pay farmers who adequately manage their lands and forests, and therefore become providers of ecosystem services (through “payment for environmental services” schemes). Eligible interventions include the maintenance of watersheds, the protection of native vegetation, tree plantation, the protection of riparian forests, soil management and protection, no-till agriculture, the restoration of degraded pastures and agroforestry systems, among others.

The theme of ecosystem services in the Cerrado biome is still dominated by ecological and academic analysis. Watanabe and Ortega [73] made a simulation analysis to quantify how relevant water and carbon are as ecosystem services provided by the Cerrado. They simulated the impacts of changes in land cover for water and carbon biogeochemical processes. The land cover change history of the Taquarizinho watershed, located in the eastern state of Mato Grosso do Sul, was used as the baseline since the region has been converted into crops and pastures since the 1960s. The impacts on ecosystem services were estimated by creating a hydro-carbon model that used several parameters related to water and carbon dynamics (evapotranspiration, channel discharge, groundwater recharge, biomass, litter, and soil). The model allowed us to simulate such dynamics for different land uses for an entire river basin, and to ordinate the values of ecosystem services associated with each vegetation cover category.

When studying the relationship between the establishment and management of agroecosystems and the provision of ecosystem services in three Brazilian biomes (Atlantic Forest, Cerrado, and Caatinga), Turetta et al. [74] assessed the use of soil parameters as indicators to monitor changes in agroecosystems. They found that water infiltration, nutrient cycling, sediment retention, and carbon sequestration and accumulation are the soil functions most affected by the agroecosystems. The authors also indicated four Brazilian public policies that present opportunities for farmers to improve the provision of ecosystem services from the agricultural system, by encouraging sustainable soil management practices. These include the following: “Plano Setorial de Mitigação e de Adaptação às Mudanças Climáticas” (Plano ABC), the Sector Plan for Mitigation and Adaptation to Climate Change for low carbon in agriculture; “Programa de Aquisição de Alimentos” (PAA), the Food Acquisition Program; “Programa Produtor de Água” (PPA), the Water Producer Program; and “Programa Nacional de Alimentação Escolar” (PNAE), the National School Feeding Program.

A more recent trend has shown conservation and economic gains when farming is in compliance with the Brazilian forest law, the Forest Code [75,76]. This analysis was carried out to demonstrate the value of changing the focus of framing ecosystem services from a project-by-project approach to a landscape-level approach. It did so because the landscape-level approach allows for a better understanding of the environmental externalities of land use, and explicitly shows the options when it is necessary to balance economic development with the conservation of natural ecosystems. The case study evaluated a commercial sugarcane expansion in a watershed in the state of Minas Gerais. The analysis applied economic and biophysical models to quantify the benefits of the Forest Code under different compliance scenarios, either at the farm level or the landscape–watershed level. For both scenarios, the best planning for habitat protection and forest restoration was evaluated.

The authors found that compliance with the Forest Code imposes low costs on businesses and can generate significant benefits for conservation by supporting additional species, storing additional carbon, and improving surface water quality. Landscape level compliance was shown to reduce total business costs (that ranged between USD 19 million and up to USD 35 million) for the period of a 6-year sugarcane growing cycle, while supporting more species and more carbon. The analysis showed that it is possible to maximize net returns from sugarcane production at the landscape level, and at the same time meet the requirements of the Forest Code.

Mapping ecosystem services at the landscape level has also been shown to be useful. Researchers developed a comprehensive approach to map ecosystem services in the Cerrado based on a spatially explicit approach, called MapES [77]. The approach is based on the premise that knowledge can be applied into decision-making related to land use and to protect the capacity of ecosystems in providing environmental services. Knowledge about eight ecosystem services (erosion and runoff control, water supply and quality, soil quality, conservation of biodiversity, food production, and energy production) was structured to provide parameters that can be evaluated against the dominant land cover/land use within a landscape. Landscape biophysical properties are made spatially explicit on maps using information about soil types, slope, and distance to the river network. The method was then tested based on a reference map of potential natural vegetation and land uses for a meso-scale catchment (Sarandi, a 32.7 km^2^ catchment located at Brasília/Federal District).

Using these approach, users and practitioners can visualize changes from human intervention on the landscapes and assess potential impacts from land use and land cover changes on ecosystem services. The integrated assessment of ecosystem services allows users to analyze ranges of uncertainties (upper and lower thresholds) and the interrelated effects of different land uses. For instance, high values of uncertainty and wide ranges between different land uses were found for biodiversity conservation in the Sarandi catchment (uncertainty factor of 0.4; ecosystem service values between 60 and 100), which results from the uncertainty in measuring relevant criteria for this ecosystem service. Meanwhile, water supply carries less uncertainty and a high accuracy of measurements (0.05; 59 and 65).

The study points out that the most important limitation of its use is the limited knowledge of the complex interaction of land use/land cover changes and ecosystem services in a landscape unit. However, the approach can be adapted to real case situations, and can be an important tool to support decision-making and integrate ecosystem services in landscape planning.

## 3. Fire as an Ecological and Anthropological Factor in the Cerrado Biome

Long before scientists described the use of “cleaning fires” in Brazil [78], naturalists such as Warming in 1778 described the use of fire during the dry season (now known as a controlled fire), whereby the main goal was to “clean” the land or to renew pasturelands [79]. The most likely primary reason for the use of fire is that it is a cheap and easy-to-use tool. Other human groups use fire for different goals, such as the indigenous people who use it to stimulate flower and fruit production and as a hunting technique, and traditional communities who use it to clean pasture and open new agricultural areas.

Nowadays, 8% of the Cerrado biome burns annually, which represents 17 Mha and occurs across all types of vegetation or phyto-physiognomies [80]. Cerrado’s natural fires are caused by lightning, and mostly occur at the transition between the dry and the wet season (October to November), when lightning without rain occurs, or during dry spells (“veranicos”) in the wet season. Human-made fires occur mainly during dry season (July to October). While some plants appear adapted to fire, others suffer irreversible damage if the frequency of the fire changes. Fire also leads to changes in the vegetation’s structure (trees’ height and density), and the recovery of the vegetation can take decades.

No doubt fire plays a role in the native Cerrado by transforming the vegetation structure, species composition and dynamics [81,82,83], and is one of the most important determinant factors, especially when used to maintain the diversity of phyto-physiognomies and biodiversity, or used by humans as a management tool to open areas for agriculture and for keeping native pasture green at the end of the dry season.

Nonetheless, there is still no full comprehension about how fire affects the Cerrado vegetation or even what is the adequate fire regime (frequency, timing, and severity). So, there is a conflicting approach to policy-making regarding the use of fire, which ranges from full protection against the use of fire to the proposal to use fire as a management tool to reduce fuel biomass and avoid large late wildfires.

### 3.1. Fire and Vegetation Formations

The Cerrado biome can be described as composed of two major vegetation formations: savannas and forests. The characteristics used to differentiate savannas from forests include vegetation structure (trees height and density), the dominant life forms (trees, shrubs, herbs, grasses), and phenology (deciduous—when trees and shrubs shed leaves). Savannas are the dominant form in the Cerrado biome (~75%) and contain two vegetation strata, one layer with scattered shrubs and trees and another with grasses or graminoids that dominate the herbaceous layer; on the other hand, forests (~25% of the biome) are dominated by trees with a closed canopy of at least 70% in the rainy season but with no grass layer [23,84]. Because of these differences, savannas and forests show contrasting fire behavior and effects.

### 3.2. Fire Behavior

#### 3.2.1. Forest Formations

There is little information of the effect of fire on forest phyto-physiognomies in the Cerrado. In general, the forest formations present a microclimate, near the surface of the soil, that is unfavorable to the propagation of surface fire, such as high humidity, lower temperature, and the absence of wind [85], as well as the presence of a litter layer or organic matter on the ground, but not a herbaceous layer.

The occurrence of natural fire in forest formations is rare; however, surface fires do occur and consume the litter deposited on the ground, causing death and/or top-kill (destruction of the above-ground structure of the plants) of some plants. The flame height is about 40 cm, but does not reach the canopy of the vegetation [86,87,88,89]. Only 2.8% of the forest area burned annually is the result of surface fires [87]. Balch et al. [90] reported that surface fire in a forest is not trivial, as in their study it took four consecutive days, with the addition of kerosene and fuel biomass at some points, to perform a prescribed burning in the experimental areas in a forest in the transition zone within the Cerrado, even after more than two months without rainfall.

Most of the burning in forest formations occurs in previously deforested or altered areas [91,92,93]. In general, burning occurs where vegetation has been cut previously and then burned (slash and burn) or in partially deforested areas, when slashing the vegetation is not necessary [94]. Slash and burn is essential for the formation of pastures or agricultural lands in the Amazon [85].

#### 3.2.2. Savanna Formations

Due to the structural characteristics of the savanna vegetation, that is, a continuous herbaceous layer with great variation in the density of woody plants that do not form a continuous canopy, and the seasonality in rainfall, the fires in savannas are usually surface fires. The fuel biomass, where fire propagates, is in the herbaceous layer, which is composed mainly of grasses or graminoids, leaves and stems (diameter smaller than 6 mm). The fire is fast—fire spread rates between 0.2 m/s and 0.5 m/s are usual, but sometimes in very special environmental conditions they could reach a maximum value of 2.2 m/s [95,96] and the mean flame height reaches 2.5 m, sometimes scorching some crowns of arboreal individuals [95,97,98,99].

### 3.3. Cerrado Annually Burned Area

The Brazilian Space Agency, INPE, has monitored changes in the Amazon forest cover since 1988. Starting in 2016, INPE also monitors changes in Cerrado cover. In Brazil, the total area burned annually is around 35 Mha, of which over half is in the Cerrado, or 17 Mha (Table 1). Rosan et al. [92] estimate 16 Mha/yr of Cerrado burned area from 2001 to 2015, where 52% of the area presented no fire during the period, 35% with two fires, 10% with five fires and 3% more than five fires.

We used data for the 2010 fires not associated with deforestation, whereby of the total area burned, 94.2% was savanna and 5.8% forest, and this calculation considered the park savanna, woody grass savanna, wooded savanna, forested savanna, wooded steppe savanna, forested steppe savanna and fluvial and/or lacustric (riparian forest) which influenced vegetation as savanna formations. Of the burned savanna, 76% was wooded savanna and park savanna. Of the burned forests, 98% were seasonal forest (Appendix II of the Third National Communication of Brazil to the United Nations Framework Convention on Climate Change—MCTI 2016 [100]).

Natural fires are ignited by lightning (mainly during the transition from dry to wet season). There is no lightning during the dry season (May to October), therefore dry season fires are due to the increased role of humans since indigenous occupation. Anthropic fires are generated especially by farmers for renewing pastures, opening new areas for agriculture, or even controlling pests. Rainy season fires, on the other hand, may have either natural or anthropic origins [101,102,103].

Recently, some experts have considered that fire exclusion from the Cerrado would be by itself a “disturbing agent” of ecosystems. Therefore, there has been proposed the implementation of Integrated Adaptive Fire Management (MIF) in protected areas with the goals of reducing fuel biomass (and consequently the occurrence of severe fires), controlling invasive grasses, or allowing for the use of fire by traditional or indigenous communities [104,105,106,107,108].

### 3.4. Fire Frequency in Phyto-Physiognomies

Experts have reported the frequency of occurrence of fires in protected areas in order to understand fire return. In the Cerrado’s conservation units, Alvarado et al. [109] observed that for 80% of burnings in the period from 1984 to 2014, the interval between burnings was less than 5 years. Pereira et al. [110] reported for savanna’s open physiognomies, such as park savanna and wooded savanna, that the burning frequency is every 3 years, and together these studies represent ~84% of the area burned annually. Denser phyto-physiognomies, such as forest savanna, present a fire interval of 5 years and represent about 4% of the area burned annually. Vegetation associated with drainage, such as “vereda” palm forest with graminoids, present intermediate values of fire return (3 to 4 years).

### 3.5. Vegetation and Species Adaptation to Fire

European naturalists exploring the Brazilian landscapes in the early and mid-19th century (for instance Langsdorff and Warming) already described the recovery of Cerrado vegetation after fire and the association between edaphic and climate aspects to explain fire frequency, the plant’s architecture, and vegetation density [79,111,112]. In the early 1960s a series of studies conducted by botanists associated the existence of xylopodium and other underground organs as features that allowed the recovery of plants after fire. Today, it is known that there is a large diversity in the underground systems in the Cerrado plants from which originate the belowground buds (so called “bud banks”) that allow re-sprouting after fire [113].

For a long time, these belowground features, and thick barks of shrubs and trees, have been considered to have evolved as adaptations to drought and/or nutrient-poor Cerrado soils. More recently, it has been reported that they should be considered as fire adaptations since these attributes coincided with the expansion of tropical savannas grasses [114,115,116].

Even though Cerrado plants may have these traits, fire can still affect species distribution and the composition of species in the ecosystem. Some aerial structures can be consumed during fire, diminishing plant biomass, and damaging or even killing the plant. As such, those adaptative traits should be useful for protecting or allowing the persistence of species, but that depends on fire regimes, especially how frequently burning occurs. It is known that increased fire frequency can diminish re-sprouting capacity, even of adapted species, and those species with no such traits (not adapted) may suffer increased mortality [117,118].

### 3.6. The Importance of Long-Term Ecological Studies

The Cerrado contains one of the richest floras in the world—more than 12,000 plant species have been described, of which 80% are herbs and shrubs found particularly in open savannas [119]. At least 40% of this flora is considered endemic to the Cerrado [120]. New plant species endemic to the Cerrado have been recently described in the literature, such as Poaceae [121], Apocynaceae [122], Convolvulaceae [123], Lentibulariaceae [124], Orchidaceae [125], and Leguminosae [126]. In addition to endemism, Ratter et al. [127] indicate that the distribution of most woody species is rare, that is, 682 species of the 914 inventoried occur in 50 or less studied sites, with 334 species found in a single site.

Even though fire in the Cerrado has been known for a long time, it is only recently that researchers started to study its effects on the vegetation and ecosystems. Up to 1990, there were no fire experiments, and reports were primarily about accidental fires on previously established research plots or short-term observations on species composition or flowering. From long-term studies of fire on Cerrado vegetation conducted in Brasilia, it has been possible to monitor the effects of more frequent fire events in the same site, and what modifications this brings to the structure and functions of ecosystems, especially the role of more frequent anthropogenic fires set during dry season.

### 3.7. How Fire Affect These Two Savanna’s Strata: The Dynamic of Recovery after Fire

#### 3.7.1. Herbaceous Layer Vegetation

Fire directly affects herbaceous layer species, since some plants are totally consumed by fire, such as grasses and other graminoids (mostly of monocots as Cyperaceae, Xyridaceae, Iridaceae), while some are only partially consumed by fire (eucots). It is possible to identify four patterns of re-sprouting after fire [128], based on when maximum biomass is reached: (1) at the beginning of wet season; (2) in the middle of the wet season; (3) in the second half of the wet season; and (4) end of dry season.

Several studies accounted for the number of herbaceous layer species flowering after fire. For instance, Coutinho [129] recorded massive flowering a few days or weeks after fire (Figure 3). Silva and Nogueira [130] recorded 96 species flowering one month after a fire and 147 species one year after the fire. Munhoz and Felfili [131] recorded 106 species flowering 3 months after a fire and 128 species eight months after it.

Regardless of the pattern presented by the species, one year after a fire at least half of fuel biomass (mainly grass and graminoids) is recovered, depending on the fire frequency, the timing when the fire is set, the precipitation, and the type of soil. In the central region of the Cerrado biome, at least 70% of fuel biomass is recovered until the middle of the following rainy season. Dead biomass represents 65% of the total, and this percentage is quite important information, for it suggests the possibility of sustaining a new fire event later on if there is enough fuel biomass [132].

Knowing the dynamics of the fuel biomass allows us to understand if or when Cerrado can be burned, and infer that open savannas physiognomies, such as the wooded savanna, the woody grass savanna, the park savanna, and the fluvial and/or lacustric influenced vegetation, can potentially be burned anytime one year after a previous fire.

#### 3.7.2. Woody Layer

In general, the woody layer is not consumed by fire, but high temperatures can damage parts of trees, and their recovery depends on the frequency of fire and fire season. Fire adaptations, such as thick bark, that allow resistance to high temperatures during fire, or the regrowth ability after a fire that increases the persistence of the species, are important traits, since high temperatures can directly affect plants, especially small plants and trunks, or indirectly affect them through hot air (convection air), resulting in damage to branches, leaves, flowers, and fruits. The main types of damage and resprouting (Figure 4) are described by Sato et al. [133]:Light damage—partial damage with aerial resprout (epigeal), that is, new leaves and branches on the main trunk;Moderate damage or top-kill—partial damage with basal and/or subterranean resprout (hypogeal) and death of main trunk;Severe damage—total damage with no resprout after at least two growth seasons, that is, plant mortality;Top-kill + Severe damage = stems destroyed.

Thus, the size of the plant is important; not only plant height that may allow plants to escape the flame zone, but also trunk diameter, because there is a positive correlation between diameter and bark thickness. Tree seedlings suffer severe damage (50%) or top-kill [134]. Most small woody plants, such as subshrubs and shrubs (less than 5 cm in basal diameter), could suffer from moderate to severe damage. It has been reported for instance that for a park savanna, 22.5% of woody individuals (diameters equal or larger than 2 cm) died after a prescribed fire [135]. More than 60% of alive plants presented light damage after the first fire; after the third fire, this was reduced to 42%. The number of sprouting plants also diminished, decreasing the capacity of individuals to survive. Fire frequency therefore is an important indicator of savanna vegetation becoming more open.

Damage to woody plants (5 cm or larger in diameter) submitted to biennial prescribed fires in park savanna and wooded savanna was reported by several studies [133,136,137]. Severe damage, after a first fire, was less than 13%, ranging from 5% for park savanna burned in August to 13% for wooded savanna burned in September. The severity of the damage increases after five biennial fires, reaching up to 48% (late burn on park savanna). The timing of the fire appears to be an important factor that leads to changes in vegetation structure, whereby late dry season fires can cause more destroyed stems (75% on wooded savanna) than early fires (41% on park savanna). Even with these high percentages of stems destroyed, some plants reach the minimum diameter size (5 cm or more—new individuals). Counting the plants that reach the minimum diameter size along with plants that suffer only light damage results in the maintenance of at least 65% of trees for early fires and only 28% for late fires. So, in addition to fire frequency, the timing of fire onset (the fire season) is indeed an important factor that affects arboreal individuals, changing its architecture, population size and reproductive factors [138,139,140].

### 3.8. Changes in Vegetation Structure

Fire frequency and fire season, then, are important factors in changing the structure of the vegetation. Late fires reduce basal area and tree cover by destroying the stems and favoring the herbaceous layer occupation of the ecosystem, especially of grasses [137,141]. In contrast, fire exclusion may increase woody individuals and fire-sensitive species, as shown for forested savanna and park savanna structures, so the vegetation becomes more closed without fire [141]. Nevertheless, despite those increments, there is no evidence that fire changes one phyto-physiognomy into another entirely, since edaphic limitations play an important role in determining Cerrado physiognomies [142]. Only when the vegetation before fires was already a more closed physiognomy can the increment of woody plants and tree species, after a long time of fire suppression (over 20 years), change phyto-physiognomies [143,144,145].

### 3.9. Ecosystem Functions: Water and Carbon Exchange between Ecosystems and the Atmosphere

Changes in vegetation structure and composition alter the functioning of ecosystems [11,12,146]. For a park savanna, after three mid-biennial fires, there was an increase in the evaporation from 0.5 to 1.0 mm of water per day, and a reduced use of deep water in the soil (below 1 m) [146]. In an unburned wooded savanna, there was a slightly larger demand for water than frequently burned savanna; in this case, the burned savanna uptakes water mainly from the top 2 m [147].

Carbon fluxes are also related to changes in vegetation structure and composition. In an unburned park savanna, the carbon accumulation was 2.9 MgC/ha, while for a burned park savanna it was 4.6 MgC/ha [12]; for wooded savanna, the carbon accumulation was between 1.5 MgC/ha and 2.6 MgC/ha [148,149,150]. In open savannas, where grasses with the C_4_ photosynthesis type are the major component of the total biomass, the carbon accumulation is as high as that reported for burned park savanna.

### 3.10. Fire and Conservation

Fires are natural disturbance events that maintain savannas as mosaic vegetation from grassland to woody savanna. This depends on a combination of the fire regime, especially fire frequency and season, the soil (type, depth, nutrients content), and precipitation. Some experts called this the “vegetation dynamic alternative stable state” or “savanna stable state theory” [151,152,153]. Though this concept has value for expanding our scientific knowledge on Cerrado ecology, it is still unclear how it can become a more operational concept for Cerrado conservation or fire management.

Regardless of whether it is necessary or not, Cerrado’s fire is always an issue or a threat for decision-makers, mainly because burning releases carbon into the atmosphere, and kills trees and animals. Some recent research discussed whether or not fire should be used and how to change policies from zero tolerance to integrated fire management [154,155,156,157].

In the Brazilian policy-making regime, the conservation of biodiversity and ecosystems can be achieved by two major categories—by the public and the private sectors. Public conservation interests are established by the designation of protected areas (conservation units in the Brazilian jargon) that can be local, state, or national in scope. They are of different sizes and goals, but usually are established for the conservation of landscapes, biodiversity, and ecosystem services. Other public lands, such as indigenous reserves and quilombolas communities (traditional slave descendants), also support conservation, but have different goals such as keeping integrity, the perennialism of their culture, and the untouchability of traditional populations.

Conservation in private lands, on the other hand, is either mandatory as per the Forest Code by which farmers have to comply with setting aside lands for “legal reserves” and “permanent protected areas”, or voluntarily designated set asides, by which farmers or land owners can create private reserves, usually in perpetuity following the normative of the national systems of protected areas (private reserves for nature preservation, or RPPN in the Brazilian jargon).

Usually for both public and private conservation, a major concern is the restoration of degraded lands and how to manage fire as a tool to ensure biodiversity, landscape, and ecosystem services. For the Cerrado, fire is already explicitly considered in the policy-making framework, as shown below:Within the Forest Code, the Ministry of the Environment submitted a bill in 2018 to regulate the use and control of fire in rural areas, and the definition of governance and roles for the Federal and state agencies. The bill includes the implementation of integrated fire management approaches, which have been used already in some protected areas, indigenous lands, and private lands throughout the Cerrado. The bill is under analysis by the Brazilian National Congress.Within the Action Plan for Prevention and Controlling of Deforestation and Fires (PPCerrado, one of the instruments of the National Policy on Climate Change), the prevention and control of forest fires is one of its main objectives. Already, the Cerrado Monitoring Programs (Prodes and DETER/Cerrado—http://terrabrasilis.dpi.inpe.br/) provide information on forest fire risks and the contribution from deforestation and fires in the Cerrado to the national greenhouse gas emissions inventories.

## 4. Agriculture Development and Changes in Land Use in the Cerrado

Though human presence in Central Brazil dates back more than 9000 years, it is only in recent times that human occupation has occurred. Since colonial times, Brazilian society has thought of central Brazil as a region to be conquered and transformed. The first permanent settlements were established by the Portuguese in the early 18th century and were associated with gold mining [5]. After the Paraguay War (1864–1870), Cerrado occupation was promoted by Brazilian authorities, who were concerned that the Cerrado’s exceptionally low population density meant the country would be unable to defend and maintain its border.

The first major economic boom in the Cerrado came during the period 1920–1930, when coffee growing and processing industries were booming in the state of São Paulo, which then became the major market for Cerrado cattle. From 1930 through 1945, the Federal government actively promoted the colonization of southern Goiás, providing land, subsidies, and technical assistance, and encouraging farmers to settle on and clear relatively fertile forested lands [3,158]. This had important implications for the Cerrado’s land use. Until the late 1950s, the contribution of the Cerrado to Brazil’s agricultural output was very low, contributing less than 10% of the national total. Dramatic changes started in the late 1960s, and by the 1970s and 1980s, the Cerrado became Brazil’s major producer and exporter of important cash crops and beef.

The major obstacle to the occupation and development of the Cerrado has been its distance from the major Brazilian urban areas along the coast and the lack of a transport system. Construction of the first railway was initiated in 1905, and after 1946 roads replaced railways as the main means to link the Brazilian regions. The construction of Brasília in the 1960s at the core of the Cerrado made possible the linking of the new capital by highways to the main Brazilian cities, and brought the modern occupation of the Cerrado [5].

Today, new links to connect the Cerrado to external markets are planned or underway. Most noteworthy among these are the paving of the BR-163 highway in central Amazonia to carry soybeans to the international port of Santarém in the Amazon river and the “Pacific” highway that connects the Cerrado to the Pacific Ocean through the state of Acre, Bolivia, and Peru. The construction of railways is again becoming fashionable. One of the most notable projects is Ferro Norte, a railway that is going to link the Cerrado to Brazil’s largest port (Santos), in order to carry soybeans to external markets.

One of the main instruments used by the Brazilian government to promote agricultural development during the 1960–1980s was subsidized credit, particularly low-interest farm loans [5,6]. Subsidies were also granted for fertilizers, pesticides, and machinery. As these incentives were offered at fixed rates, lower than the high inflation rates of the time, they all represented further subsidies to producers. As a result, credit to the rural sector between 1969 and 1979 was 199% for agriculture and 164% for livestock production [5]. After 1979, credit flows slowed, especially long-term investment credit.

Loans were not evenly distributed among crop types. Over 75% of production loans were concentrated in six crops: soybeans, rice, coffee, wheat, maize, and sugarcane. Soybeans alone received 20% of the credit available to Brazilian farmers [5]. Since loans were allocated based on the area planted, it encouraged extensive and inefficient agriculture [159]. Cerrado farmers received 70% more credit than Amazon and northeast farmers, where the small producers of food crops are concentrated. This readily available credit increased the demand for land in the Cerrado and drove up land prices [5,6].

In addition to the incentives and subsidies policies, other strategies have been used to develop and occupy Cerrado land, most importantly regional developmental programs, the deployment of new technologies, and minimum prices policies.

The Program for the Development of the Cerrado (POLOCENTRO) of 1975, a public program, settled farmers in places with good farming potential, improved secondary roads and electricity, and developed agriculture research and technology [5]. The program had a major impact on Cerrado agriculture. Big farms (larger than 1000 hectares in size) accounted for 39% of all projects and received more than 60% of the credit. The program’s original goal of giving preference to foodstuffs was never realized, and, instead, the program induced the expansion of commercial agriculture (cattle ranching and soybeans).

POLOCENTRO was closed in 1985, but already in the late 1979s, the Brazil–Japan Cooperative Program for the Development of the Cerrado (PRODECER), a private program, was created. PRODECER selected experienced farmers from the south and southeast of Brazil to be settled in the Cerrado for soy production. It was financed by loans from both the Brazilian and the Japanese governments, but contrary to the POLOCENTRO, loans were granted at real (not fixed) interest rates. PRODECER ended in 2001.

The creation in 1973 of EMBRAPA (the Brazilian Agricultural Research Corporation, a public company) and its several decentralized research centers allowed for the development of appropriate technologies to enable farmers to deal with Cerrado’s acidic soils with low nutrient availability, and the tropical climate (rain seasonality, temperature, humidity, evapotranspiration rate, wind), as well as genetic improvement, has also promoted agricultural expansion. The technologies included the use of modern machinery in agricultural operations, application of phosphate fertilizer and lime to correct for soil nutrient deficiency and acidity, Rhizobium-based nitrogen fixation, the development of crop varieties, the use of herbicides and pesticides, and modern machinery [3,160].

The price support policy in Brazil, which guarantees a minimum price for agricultural products, has been in effect since the 1930s. In the 1980s, farm credit lines were restricted and reduced or eliminated. Consequently, production costs increased substantially in the Cerrado. The government then started to purchase large amounts of Cerrado products, particularly soybeans, rice, and corn [6]. The net result was the expansion of commercial farming in areas that could not have supported profitable production without the subsidies and, consequently, the deforestation of new land.

## 5. Evolution of Cerrado’s Policy Framework

In the past, Brazil’s land use policies were often formulated with little attention to their implications for land degradation, social conflict, and biodiversity conservation. However, over the past 15 years, Brazil has built a robust set of environmental regulations at the national and subnational levels [72,161], yet few of these policies are specific to the Cerrado’s conservation and sustainable use. Priority and attention have been given to the Amazon forest, not the Cerrado. Therefore, the Cerrado is underrepresented in the National System of Conservation Units (SNUC); 8.7% of the Cerrado’s surface area is under legal conservation compared to 26.1% of the Amazon’s [162].

The Brazilian Forest Code demands farmers to conserve 20–35% of their properties for sustainable use and conservation purposes. The implementation of the Forest Code, which applies to both the Amazon and the Cerrado, has emerged as the most important institutional driver of land use in the biome (a combination of agriculture production and ecosystem conservation).

However, the Federal and the state-level governments are still struggling to implement the law, especially due to (i) low institutional capacity at the regional and local scale, (ii) persistent problems with land conflicts and land tenure regularization, (iii) a lack of incentives to farmers to fully engage them in the Forest Code implementation, and (iv) a recurring change in the regulatory framework related to Forest Code implementation terms.

Despite these factors, the deforestation rates in the Cerrado have decreased from around 30,000 km^2^ for the year 2001 to 6900 km^2^ on average for the last three years [163,164]. For the methodological and regional differences in deforestation rates, see Alencar et al. [61]. There is an increasing pressure from civil society and the private sector to implement more sustainable practices in the biome. Introducing the next generation of monitoring systems and improving the transparency of property and land use data will be key for building on this new momentum.

Land use and ecosystems are key for achieving the Sustainable Development Goals (SDGs) and the Paris Climate Change Agreement. These goals set an ambitious roadmap to inspire government planning and the mobilization of stakeholders that recognize the three dimensions of sustainable development in a comprehensive set of goals and targets; they allow the movement from theory to practice. There is no question the COVID-19 pandemic is also exerting an extraordinary pressure for more sustainable agriculture production and green investments in Brazil [165,166].

Public investments are driving the demand for sustainability in the Cerrado, and there is a substantial portfolio of budgetary resources, national and international funds, and cooperation projects that are ready to finance sustainable land use in the biome. Farmers’ access to these resources and their transactional costs are still a barrier, but incentivizing a transition to deforestation-free supply chains, pasture and forest restoration, and territorial consolidation in the Cerrado, are anticipated outcomes.

The private sector has voiced major concerns regarding the still relatively high levels of deforestation related to the expansion of agriculture and pasture. Eighteen initiatives seeking more sustainable production have been introduced in Brazil recently, sponsored by foundations, NGOs, traders, companies, and banks to promote the economic benefits of land use that help conserve ecosystems. The implementation of these actions still poses challenges: initiatives focus mainly on supply chains (particularly of soy and beef) that aim at “zero net deforestation” in the Amazon and the Cerrado, and there is almost no support for complementary actions or the avoidance of fragmented (or antagonistic) initiatives. Another key challenge is the noticeable absence of a clear and extensive engagement with farmers.

The new political agenda of the Federal government for agriculture and the environment, and the recent free trade agreement between Mercosur and the EU, may have important impacts on land use in the Cerrado. Promoting the better coordination of policies that aim simultaneously at ecosystem conservation, sustainable land use, and climate change mitigation is still a challenge in the Cerrado. In Appendix A, we describe the most relevant policy instruments that might drive more sustainable land use in the Cerrado in the near future, as well as current funding mechanisms for advancing initiatives in the Cerrado.

## 6. Building a Better Future for Sustainable Agricultural Production in the Cerrado

There is an unprecedent opportunity to develop sustainable solutions for the Cerrado by bringing the main stakeholders together—the business sector, foundations, public policy makers, researchers, rural farmers, and civil society. The goal is to understand the trade-offs between ongoing land use transformation and the consequences to ecosystems in the Cerrado, to identify opportunities for land use decisions that maintain ecosystem functions without constraining food and bioenergy production, and to build a new narrative for ecosystem services for the Cerrado.

### 6.1. Why the Cerrado Needs a New Road Map for Production

Brazil’s abundant lands in the Cerrado and the ingenuity and hard work of its farmers and producers have transformed the nation into a global agricultural powerhouse. Today, Brazil ranks as one of the world’s leading agribusiness producers and exporters [4]. From 1977 to 2019, agricultural production skyrocketed from 8 Mt to 130 Mt. Yet to maintain Brazil’s global standing to meet the world’s food demand, reduce hunger, and rehabilitate land, a new and ambitious roadmap for production in the Cerrado is needed; one that is rooted also in environmental sustainability.

Brazil has already proven that agricultural development can continue to grow even when deforestation is curbed, if technology, policies, and science is correctly deployed. The next step is to demonstrate how ecosystem protection can add economic value to agricultural production in the Cerrado.

Our current scientific knowledge shows that Cerrado’s natural vegetation can play a fundamental role in conserving water and protecting the healthy functioning of ecosystems. This is why Brazil’s private and business sectors need to join forces with national and international academics, civil societies, nonprofit organizations, and governments to improve practices and supply the rising demand for agriculture products that have been developed sustainably.

Fortunately, Brazil is ideally positioned for creating a more sustainable business model (see MAPA 2020 [167]). The Cerrado’s business and policy environment has many strengths that already bring agribusinesses and conservationists together [168,169], including:A highly competitive agribusiness industry;A vigorous commercial pulp, paper and timber industry;The ability to increase productivity in the region significantly through improved technology;An innovative and growing economic restoration industry;An advanced research and development (R&D) capacity;Consolidated civil society organizations;The opportunity to cultivate already cleared areas and lands for agricultural production;A strong policy framework to promote ecosystem services and conservation in private lands;Evidence that shows the increased enforcement of deforestation does not undermine agricultural productivity or economic growth;Proven scientific and technical capacity to monitor ecosystems and deforestation.

This new collaboration between agribusiness and ecosystem conservation, if fully developed in the Cerrado, will create a transformative new business model for Brazil and the world. Formulating a new land use agenda will require buy-ins from both the agricultural and environmental constituencies, and must be based on solid science. If successful, Brazil has the potential to double its agricultural productivity while protecting its natural resources [161]. The new business model would establish Brazil as an example for the world and a global leader in agricultural growth and ecosystem protection.

### 6.2. Ensuring the Potential of the Biome

We propose to develop a pragmatic agenda for the Cerrado together with the agribusiness and conservation constituencies. The Cerrado has already seen many concrete examples of agribusinesses and conservationists working together successfully.

Many farmers are seeking to achieve higher agriculture productivity while also seeking compliance with environmental regulations.Think-tanks, NGOs, academics and traders are building strategies for zero-deforestation in the mid-term.Financial flows and the mobilization of private investments have increased for green investments.New technologies and artificial intelligence are being deployed for better planning of the farming landscape.Farmer awareness regarding the environmental risks associated with agricultural production has increased.

In addition, much of Brazil’s 180 Mha of pastureland is degraded with low productivity, and therefore represents a major opportunity for agriculture, bioenergy, and beef production to expand into this pastureland without further deforestation [170]. In several locations, low productivity pastureland is already being replaced by more intensive techniques and knowledge. Examples such as that described by Assunção et al. [171] for Mato Grosso do Sul underscore the potential to achieve the dual goals of improved productivity and reduced deforestation in the Cerrado.

## 7. Conclusions: Toward a Stronger Future in the Cerrado

The insufficient scientific knowledge on ecosystem services must not justify inaction. Our knowledge would be of benefit if studies sought to find synergies between changes in land use/land cover and how they affect natural ecosystems and agroecosystems. To be policy relevant, knowledge from ecosystem services must clearly specify the spatial and temporal scales from which the best results are achieved. Assessments of ecosystem services must perform quantitative predictions and specify the implications for the existing policy options.

Knowledge about ecosystem services must provide evidence that benefits stakeholders. A key for protecting the Cerrado biome is to understand how the existing policy framework (especially compliance with the Forest Code, which demands huge investments in how best to design the landscape) can support decision-making by farmers about where to conserve the natural vegetation as required by the law.

A pragmatic and conciliatory land use agenda must be based on scientific knowledge and must support innovative decision-making solutions for policy-makers and stakeholders. To make scientific knowledge policy-relevant, it must be combined with stakeholders’ and institutions’ engagement, particularly farmers and donors. Farmers will ultimately have to decide on the “design” of landscapes in the Cerrado. Donors should support this goal through the creation and dissemination of relevant ecosystem services knowledge and by working with others (e.g., the finance sector; the public sector) to stimulate the creation of incentives for farmers to make the best use of ecosystem services.

To move forward with an agribusiness landscape where the importance of ecosystems and ecosystem services are fully integrated, it will be crucial to facilitate participation among the region’s different stakeholders and organize where they fit into the new business model.

Three primary strategies must be used to build the road map that will lead to the new business plan for the Cerrado:Create an active, solution-focused dialogue among farmers and organizational stakeholders in the Cerrado. Engaging farmers will help ensure that the new solutions for improved business practices can be leveraged and scaled up. Farmers and producers must be at the table in the development of the agribusiness–ecosystem business model so that they may benefit fully from the new actions taken under the model.Develop, field test, and implement new business intelligence and tools that help farmers to manage or reduce agribusiness risk from degraded ecosystems. The development of critically needed business tools can be achieved by studying new farming practices and interviewing farmers, examining how state-of-the-art cleantechs and agtechs are operating, identifying and mobilizing new financial mechanisms and capital, and learning from new public–private approaches. The development of ecosystem indicators, environmental monitoring, economics studies, and the development of tailor-made solutions with farmers about how best to implement the Forest Code, must come in tandem as a component of this business intelligence.Formalize partnerships wherever possible that serve the development of the new agribusiness–ecosystem business model. Throughout the Cerrado, numerous organizations and funders are seeking to advance agribusiness–ecosystem ideas like these, but they often lack farmer input and coordination at the regional level.

The “design” of landscapes in the Cerrado depends on each farmer or producer’s decisions. The landscape/watershed scale seems to be the most relevant for decision-making on how to achieve production and conservation results in the Cerrado, and therefore the most promising for analytical work. The landscape/watershed scale also appears to be an appropriate level for engaging with stakeholders because at this scale, the knowledge about ecosystem services would most probably match stakeholders’ demands related to the benefits accrued by ecosystem services.

Given the status of the current debate on the role of fire in policy making, it is expected that scientists, park rangers, and policy-makers have been pursuing an agenda for the use of fire as a management tool to guarantee the maintenance of biodiversity, the integrity of landscapes, and ecosystem services. We must bear in mind, however, that there are two important conditionalities for the use of fire as a management tool:Fire frequency is a major threshold. Vegetation recovery is highly dependent on fire frequency and maintains the Cerrado’s high biodiversity, especially for the herbaceous layer, but it is a limitation to woody vegetation recovery;When combined with fire frequency, the season when fire occurs (timing) is a determinant of herbs, shrubs and trees survivorship, despite the fact that some species have useful traits (such as bud banks, underground systems and thick barks) for protection against fire that may allow for the increased survivorship of species.

## Figures and Tables

**Figure 1 plants-09-01803-f001:**
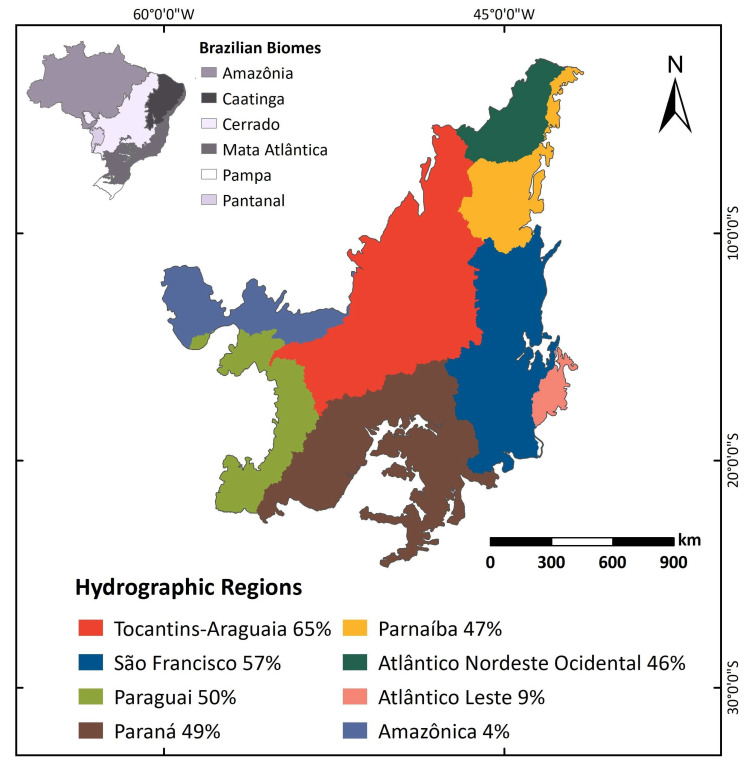
The Cerrado biome’s contribution to the hydrographic regions in Brazil (modified from Sano et al. [21]). The numbers refer to the proportion of each hydrographic basin within the Cerrado biome.

**Figure 2 plants-09-01803-f002:**
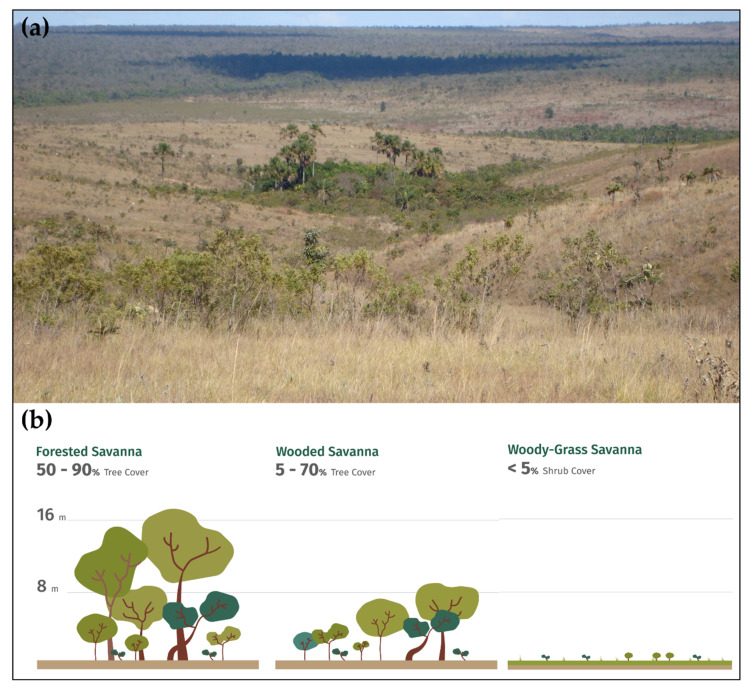
(**a**) The Cerrado’s main feature is the coexistence of trees and shrubs on a matrix of herbs and grasses. Riparian forests follow the stream and river margins (Photo is courtesy from Reserva Ecológica do Roncador, RECOR/IBGE, Brasília, Brazil); (**b**) The high species richness (number of species) and vegetation structure (height and canopy cover) shows the ecological complexity of the Cerrado.

**Figure 3 plants-09-01803-f003:**
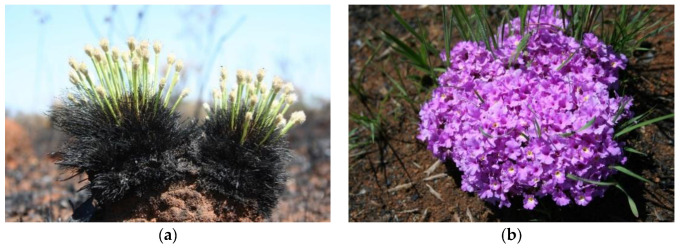
(**a**) *Bulbostylis paradoxa* flowering starts one day after fire, (**b**) *Lantana montevidensis* blooming one month after fire.

**Figure 4 plants-09-01803-f004:**
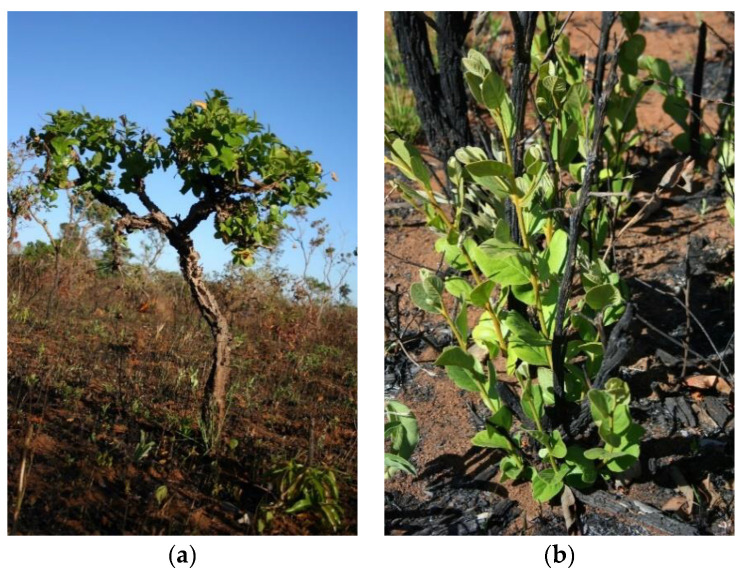
(**a**) *Symplocos rhamnifolia* with aerial resprouting (light damage), (**b**) *Styrax ferrugineus* with top-kill (moderate damage) after a fire in Cerrado.

**Table 1 plants-09-01803-t001:** Area burned annually in the Cerrado [80].

Year	Burned Area (ha)	(%)
2003	17,514,000	8.6
2004	19,868,400	9.8
2005	18,169,000	8.9
2006	11,810,900	5.8
2007	32,913,800	16.2
2008	13,945,800	6.9
2009	7,435,300	3.7
2010	30,482,500	15.0
2011	13,498,800	6.6
2012	24,745,100	12.2
2013	11,100,400	5.5
2014	17,075,600	8.4
2015	19,050,600	9.4
2016	15,114,200	7.4
2017	15,835,200	7.8
2018	8,537,400	4.2
Mean	17,318,562	8.5

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
