# Peer review of "The Role of Vegetation on the Dynamics of Water and Fire in the Cerrado Ecosystems: Implications for Management and Conservation"

_plants, 2020, doi:10.3390/plants9121803_

Round 1
Reviewer 1 Report
The manuscript (MS) under consideration is an excellent detailed literature review on a wide range of environmental and land use issues in the Cerrado region of Brazil. However, there are some circumstances that make it difficult to publish the MS in the Plant Journal:
1. The title of the manuscript does not correspond to its content. Actually this is an article on fire ecology related to agriculture and generally to land use with socio-political issues in Cerrado. It is clear from the last paragraph of Introduction: “In this paper, we reviewed our current knowledge about water ecology and fire management to show that an ecosystem services perspective can bring about a conciliation of agriculture production with conservation by supporting effective land use decision making and optimization of public policy. Both conservation of ecosystems and agriculture are of strategic importance to Brazil. We argue that the richness of Brazilian policies on land use can both provide well-managed agricultural lands and have a favorable impact on environmental conservation.”
2.The structure of MS according to lines’ account is as follow:
Introduction 6%
Vegetation 27%
Fire and Water 26%
Fire and Vegetation 7%
Agro-Socio-Political aspects 30%
Conclusion 4%
This means that only one-third of the MS is devoted to vegetation and the main focus is on ecological factors (fire and water) and agriculture combined with socio-political aspects.
3. MS has no discussion and synthesis of reviewed data for detailing “fundamental role of vegetation” declared in the title (though I am agreeing that it is true!).
In Conclusion (62 lines) word “vegetation” is mentioned 3 times only.
Therefore I have to conclude that this excellent review doesn't meet scope of the journal Plant.
I can propose the following solutions:
1. In present form, the MS can be redirected to one of these journals:
Land Degradation and Development
Landscape Ecology
Fire Ecology
Agroecology and Sustainable Food Systems
2. Authors can divide the review into two parts. One review should be about vegetation and fires, the other about agriculture, social and political aspects, but each of them should shortly mention the main points of the other review.
I wish the authors success.
Author Response
Being a review of the literature, the structure of the MS is well balanced for it combines a description of the main biological and physical features of the Cerrado savanna, the ecological role of the vegetation on the dynamics of water and fire, and the potential of using this knowledge into policy making. We do not agree that the article is about fire ecology related to agriculture.
It is not correct to say that only a third of the MS is devoted to “vegetation”. Over two thirds of the “water in the Cerrado biome” section is devoted to the description of the relationship between the vegetation and ecosystem processes, and the effects of change in vegetation cover on ecosystem services. Almost the entire “fire” section is devoted to analyzing the effects of fire on the structure and coexistence of different vegetation types, plants adaptation to fire, recovery of vegetation and species from fire, damage of fire to different types of vegetation and species, and carbon accumulation into the vegetation after fire events. Therefore, it is the entirety of the MS that supports our proposition for the “fundamental role of vegetation” into the title.
Plant water ecology and fire ecology are well developed research subjects in the Cerrado, and the MS takes advantage of this scientific knowledge to shed new light on the current challenge for Cerrado conservation that is, how to conciliate conservation of ecosystems and agriculture production simultaneously.
Finally, what our MS shows is that in order to bring the ecological knowledge of Cerrado vegetation to a broader, policy making perspective, we must also bring a balanced view of historical and current policies for the region that ultimately is to deal with the restoration and conservation of the natural vegetation. For all these reasons we believe the MS can be an important contribution to the audience of Plants.
Reviewer 2 Report
This is a very well documented original manuscript for a better knowledge of the Brazilian Cerrado territory, like a monograph. It develops several topics on policies, land management and conservation of its high biodiversity being an interesting contribution to scarce specialized scientific literature on this unique natural area.
Only two comments that should be consider as a minor revisions:
-Line 279: Pay attention to the sentence punctuation.
-Suggestion: The inclusion of several figures or pictures of the Cerrado landscapes would improve the understanding of the ecological processes described in the text (postfire regeneration, agricultural impacts, etc.).
Author Response
We have attended to all suggestions:
- Punctuation of line 279 has been corrected.
- A new figure with a photo and description of vegetation types has been added.
- We added a short new paragraph on plant endemism in the Cerrado.